# Effects of castration on atherosclerosis in Yucatan minipigs with genetic hypercholesterolemia

**Jeong T. Shim**[1,2], **Nikolaj Schmidt**[1], **Paula Nogales**[2], **Torben Larsen**[3], **Charlotte B. Sørensen**[1], **Jacob F. Bentzon**[1,2]*

**1** Department of Clinical Medicine, Heart Diseases, Aarhus University, Aarhus, Denmark, **2** Centro Nacional de Investigaciones Cardiovasculares Carlos III (CNIC), Madrid, Spain, **3** Department of Animal Science, Aarhus University, Aarhus, Denmark

* jfbentzon@clin.au.dk

**Data Availability Statement:** All relevant data are within the paper and its Supporting Information files.

## Abstract

### Background

Low plasma testosterone, either spontaneous or as a result of androgen deprivation therapy for prostate cancer, is associated with an increased risk of cardiovascular events. The underlying mechanism in humans is not understood. Experimental studies in mice have shown that castration facilitates atherogenesis and may increase signs of plaque vulnerability. Pigs used for translational atherosclerosis research have frequently been castrated for practical or commercial reasons, but the effect of castration on atherosclerosis has never been systematically evaluated in pigs.

### Objective

To study the effect of castration on atherosclerotic plaque burden and type in genetically modified minipigs with hypercholesterolemia.

### Methods

Newborn male Yucatan minipigs with transgenic overexpression of a human gain-of-function mutant of proprotein convertase subtilisin/kexin type 9 were randomized to undergo orchiectomy (n = 8) or serve as controls (n = 6). Minipigs were started on high-fat diet at 3 months of age and the amount and composition of atherosclerotic lesions were analyzed at 12 months of age. Plasma lipid profiles and behavioral parameters were also assessed.

### Results

Plasma lipids were slightly affected to a more atherogenic profile by orchiectomy, but atherosclerotic lesion size was unaltered in the LAD, thoracic aorta, abdominal aorta, and iliac arteries. The distribution of lesion types (xanthomas, pathological intimal thickening and fibroatheromas) were also not statistically different between groups in any of the examined vascular territories. The abdominal aorta developed the most advanced stages of disease with reproducible fibroatheroma formation, and here it was found that the area of necrotic

**Funding:** The study was funded by a grant from the Danish Independent Research Council (Sapere Aude Programme, 4004-00459B) and a fellowship from The Graduate School of Health, Aarhus University, Aarhus, Denmark. The CNIC is supported by the Instituto de Salud Carlos III (ISCIII), the Ministerio de Ciencia e Innovación (MCIN) and the Pro CNIC Foundation, and is a Severo Ochoa Center of Excellence (SEV-2015-0505). The funders had no role in study design, data collection and analysis, decision to publish, or preparation of the manuscript.

**Competing interests:** JFB and CBS are inventors on a patent on PCSK9 transgenic minipigs owned by Aarhus University (US 8,546,643). This does not alter our adherence to PLOS ONE policies on sharing data and materials. The other authors have no disclosures to report.

**Abbreviations:** ADT, Androgen deprivation therapy; APOE, Apolipoprotein E; AUC, Area under the curve; GnRH, Gonadotropin-Releasing Hormone; HDL, High-density lipoprotein; HFHC, High-fat high-cholesterol (diet); LAD, Left anterior descending (coronary artery); LDL, Low-density lipoprotein; LDLR, Low-density lipoprotein receptor; Orx, Orchiectomized; PCSK9, Proprotein convertase subtilisin/kexin type 9.

core was significantly increased in orchiectomized pigs compared with controls. Orchiectomy also reduced aggressive behavior.

## Conclusions

Castration does not alter the burden of atherosclerosis in hypercholesterolemic Yucatan minipigs, but may increase necrotic core area in fibroatheromas.

## Background

Low serum testosterone levels are associated with increased atherosclerosis and risk of cardiovascular mortality, both in men with spontaneous age-associated serum testosterone decline [1], and in patients receiving androgen deprivation therapy (ADT) to control prostate cancer [2]. Meta-analysis of randomized clinical trials of ADT, on the other hand, has failed to replicate the finding from observational studies and causality of the association between castration and cardiovascular disease thus remains unresolved [3].

One prevailing hypothesis is that hypogonadism may increase cardiovascular risk through atherogenesis. This has been explored in mouse and rabbit models, where castration changes fat distribution, evokes more atherogenic lipid profiles, increases atherogenesis, and leads to changes in plaque composition with increased necrosis [4–7]. Recent studies in *Apoe*-deficient mice showed that loss of testosterone signaling in thymic epithelium was responsible for part of the pro-atherogenic effect [8]. Yet it remains unclear to what extent these effects of castration in mice and rabbits may translate to other models and indeed to human atherosclerosis.

Porcine models of atherosclerosis, including wildtype and gene-modified lines, develop lesions with high pathoanatomical resemblance to human lesions [9]. Because of its routine use in pork production or to avoid unsolicited breeding of commercial minipig lines, male pigs in experimental atherosclerosis experiments have often been castrated. Importantly, however, the effect of castration on atherosclerosis in pigs has never been analyzed, and it remains unknown how castration may have contributed to atherosclerosis phenotypes reported in the literature. In the present report, we analyzed the effect of castration on plasma lipids, atherosclerosis, and animal behavior in Yucatan minipigs with genetic hypercholesterolemia.

## Methods

### Animals

Procedures involving animals were approved by The Danish Animal Experiments Inspectorate (2015-15-0201-00570). Transgenic Yucatan minipigs with hepatic overexpression of a gain-of-function mutant of human proprotein convertase subtilisin/kexin type 9 (PCSK9$^{D374Y}$) [10] were bred at the Department of Animal Science, Aarhus University. The animals were kept in a specific pathogen-free facility. Housing was initially in groups and later individually with visual, auditory, olfactory and tactile contact to other pigs.

A total of 20 male piglets were randomized to either bilateral orchiectomy performed 2–7 days after birth (n = 12) or serving as controls (n = 8). This number was estimated by a power analysis to detect a difference of 40% in aortic atherosclerosis (by en face examination) using variation estimates from a previous report [10]. Bilateral orchiectomy was performed after intramuscular injection of 5 mg flunixin meglumine per animal (a long-acting NSAID) as a pain relief. Testes were then removed through an incision through the overlying skin. Pigs

were monitored by the stable facility staff and no acute complications to the procedure were observed.

The minipigs were fed a low-fat standard pig diet until 3 months of age and subsequently a high-fat high-cholesterol (HFHC) diet consisting of standard pig feed comprising 20% (w/w) of lard and 2% cholesterol (Sigma-Aldrich) until euthanization at 12 months of age. The diet was prepared by solubilizing the crystalline cholesterol in melted lard and mixing it with a standard feed for growing pigs containing 68.0% barley, 15.0% oat, 9.6% soy bean meal, 2.0% animal fat, 3.0% molasses, and 2.4% minerals and vitamins. Feeding was ad libitum until the pigs reached a weight of 20–25 kg and was then restricted to 700 g of feed per day divided into two daily portions.

The animals were monitored daily by the facility staff throughout the study period. Six pigs were lost from the study leaving final group sizes of 8 castrated pigs and 6 controls. Three pigs (2 castrated and 1 control) were euthanized because of the occurrence of scrotal hernias. Hernias were endemic in our Yucatan minipig colony at the time, but was greatly mitigated by changing breeders suggesting that it was genetic and not a complication to the orchiectomy. Three additional pigs were lost, two euthanized because of sickness (gastroenteritis) and one dead from an unknown cause.

## Assessment of aggressive behavior

At 11 months of age, minipigs were transferred to single-pens in another stall with only males. Aggressive behavior in this setting was assessed by scoring 10 aggression parameters (yes/no scoring yielding a score from 0–10) defined in collaboration with the animal facility staff: Attempting to bite observer, biting bars on fence, pushing the observer, bumping into bars on fence, standing with front legs on top of fence, chewing noisily, digging with front legs, standing very upright, facing laterally to the observer, resists being pushed. The minipigs were also assessed for shyness of the observer (keeping a distance), which was considered a sign of non-aggression. Pigs were familiar with the observers, which were part of the daily animal staff caring for the pigs. Each minipig was assessed 1–4 times. Blinding was not possible because the lack of testes is obvious in orchiectomized pigs. An average score on each parameter was calculated for animals observed more than once. The overall total score of all aggression parameters was compared between castrated and control minipigs. Likewise, score on the non-aggression parameter, shyness of the observer, was compared between groups.

## Plasma analysis

Blood was drawn in the morning from individual animals after an overnight fast and EDTA-plasma analyzed for total cholesterol, LDL cholesterol (direct method), HDL cholesterol (direct method), and triglycerides by standard procedures using an autoanalyzer ADVIA 1800® Chemistry System (Siemens Corporation, Tarrytown, NY 10591, USA). *Intra-* and *inter* assay precision were in all instances below 3 and 4 CV%, respectively. Plasma testosterone was analyzed in the blood samples collected at 3, 6, 9, and 12 months of age by immuno-chemical methods (Arbor Assays, K032-H1, Michigan 48108 USA). The instructions given by the manufacturer were followed. Coefficient of variation between replicates were in all instances below 12%, the limit of detection was confirmed to be at least 40 pg/mL.

## Pathology

At the age of 12 months, the minipigs were sedated by intramuscular injection of midazolam (1 mg/kg) and azaperone (8mg/kg). 10.000 IU heparin was injected intravenously and animals were euthanized by a lethal dose of pentobarbital (4 mL/kg). Hearts were excised and

immersed in 4% phosphate-buffered formaldehyde. The aortas were cut transversally between the branching of the 6th and 7th intercostal arteries to divide aortas into the thoracic and abdominal part. The two aortic specimens and the proximal 8 cm of the right iliac artery were cut open longitudinally and fixed on a polystyrene plate with needles before immersion in formaldehyde for 24 hours. All specimens were then kept in cold phosphate-buffered saline.

Aortas and right iliac arteries were stained with Sudan IV (Sigma-Aldrich; 5 g/liter in 96% ethanol in 5 minutes followed by 90 seconds washout in 96% ethanol) and *en face* images of the vessels were obtained using a digital scanner (Epson Perfection V600 Photo, Seiko Epson Corporation, Japan). In thoracic aortas and iliac arteries, the area fraction of intimal surface covered with lesions was determined by automated computer-assisted measurement of Sudan-IV stained (sudanophilic) intima using ImageJ 1.48v (National Institutes of Health, USA). Lesions in the abdominal aorta were traced manually using ImageJ since most of the raised lesions in this region were less sudanophilic.

For histological examination, left anterior descending (LAD) coronary arteries were excised from the myocardium and the proximal 3 cm were divided into six 5 mm segments. The most advanced lesion (by macroscopic inspection) in two regions of the abdominal aorta were also obtained. Lesions develop reproducibly in the distal aorta near the aortic trifurcation and a cross-sectional slice containing the most raised lesion in this region was selected. Furthermore, a cross-sectional slice containing the most raised lesion furthermore proximal in the abdominal aorta (typically near the renal arteries) were obtained. Segments were paraffin-embedded, sectioned for histological analysis, and stained with elastin-trichrome for morphometric analysis and morphological measurement. Plaque necrosis, defined as cell- and collagen-free areas in trichrome and Sirius Red-stained sections, was determined in abdominal aortic lesions.

Lesion and necrotic size were measured using ImageJ (NIH). For aortic plaques, maximum plaque-thickness was measured from the internal elastic lamina to the lumen. All microscopic analyses were performed blinded. Lesions were categorized according to the Virmani classification as normal, xanthoma, pathological intimal thickening or fibroatheroma [11].

## Statistics

All statistical analyses were performed using Prism 6 (GraphPad Software Inc). Data are expressed as mean±SEM. The experimental units in all statistical comparisons were single animals. P-values were calculated using 2-tailed unpaired Student's t-test (with Welch's correction as indicated in figure legends) for normally distributed data (Shapiro-Wilk normality test). Area under the curve (AUC) was calculated by the trapezoidal method and compared between groups to test for differences in time serial measurements. For comparison of plaque types between groups, the most advanced plaque type recorded in each pig and arterial bed was compared among groups using Fisher's exact test. $P < 0.05$ was considered significant. Significant differences are marked with * in figures. Exact values for $P < 0.05$ and $P = 0.05$–0.10 are given in text and in figure legends.

## Results

Fourteen male PCSK9D374Y Yucatan minipigs were, by randomization, either orchiectomized (Orx) (n = 8) or served as controls (n = 6). Plasma testosterone was measured on blood samples collected at 3, 6, 9, and 12 months of age, and showed the expected increase around sexual maturation in intact minipigs. Orchiectomy reduced testosterone to castrate levels similar to those reached by ADT in humans (<0.2–0.5 ng/ml) (Fig 1A). Mean final body weights did not differ between groups (79.0±3.06 kg versus 77.3±3.83 kg), but orchiectomized males on

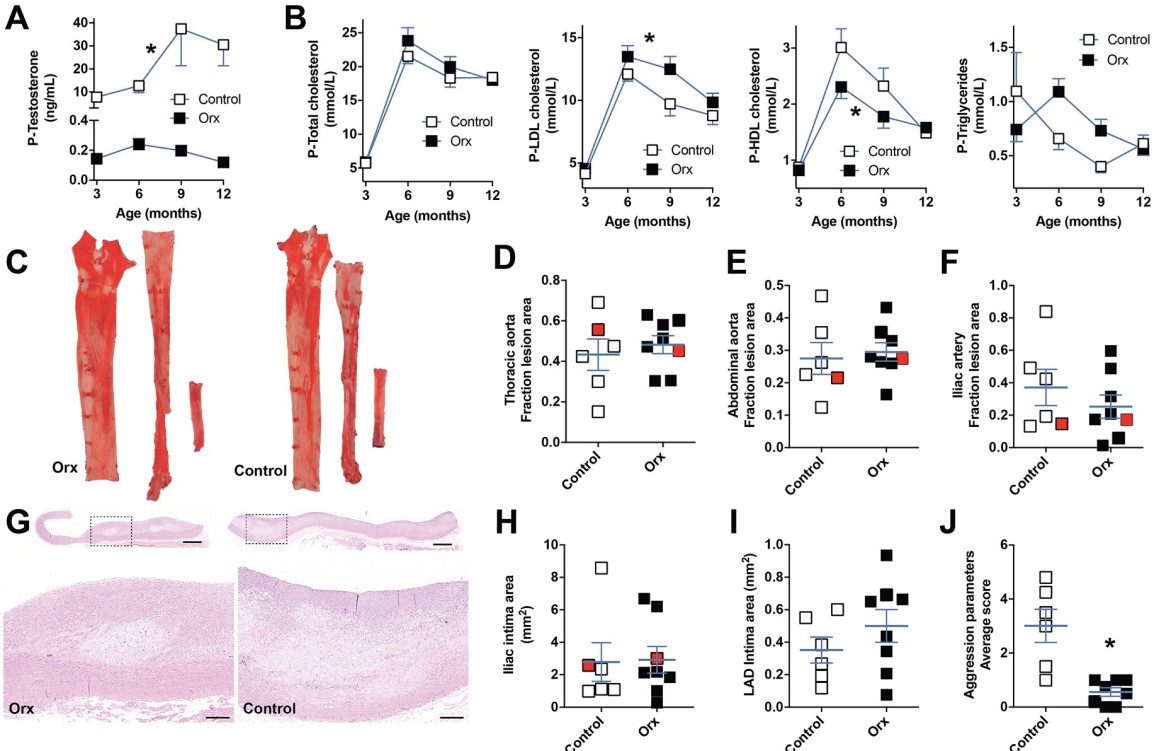

**Fig 1. Plasma measurements, plaque burden, and animal behavior. A**. Plasma testosterone levels in orchiectomized (Orx) and control PCSK9$^{D374Y}$ Yucatan minipigs. *P<0.0001 (t test on AUC). **B**. Plasma lipid levels on low-fat diet (3 months of age) and on HFHC diet (6, 9, and 12 months of age) was altered in castrated pigs with modestly increased LDL cholesterol (*P = 0.045, t test with Welch's correction on AUC) and reduced HDL cholesterol (*P = 0.002, t test on AUC). The difference in plasma triglyceride burden over the course of study was not significant (P = 0.06, t test on AUC). **C**. Representative examples of Sudan red stained thoracic aorta (left), abdominal aorta (middle) and right iliac (right). **D-F**. Quantification of lesions as fraction of total vessel surface in thoracic aortas (D), abdominal aortas (E), and right iliac artery (F) showing no significant differences between groups. Data from the examples in (C) are marked in red. **G**. Representative examples of histology of iliac lesions stained with hematoxylin-eosin. Scale bars = 500 μm. **H-I**. Intimal/lesion areas measured on hematoxylin-eosin stained sections in right iliacs and LADs show no differences between groups. Data from the examples in (G) are marked in red. **J**. Scoring of aggressive behavior showed significant reduction in orchiectomized pigs. Each animal was observed one to four times. *P = 0.009 (t test with Welch's correction). Number of data points/animals in each panel are 6 and 8 for the control and orchiectomized groups, respectively. Error bars indicate mean and SEM.

average had thicker backfat compared with controls (69.1±3.49 mm versus 49.5±7.81 mm, P = 0.027).

Plasma lipids were unaltered by orchiectomy at baseline when pigs were fed standard low-fat diet, but orchiectomized pigs developed higher plasma LDL cholesterol and triglycerides and lower plasma HDL cholesterol on high-fat, high-cholesterol diet over the course of study (Fig 1B).

## Castration does not alter plaque burden

Atherosclerosis burden was evaluated by *en face* analysis of the thoracic aorta, abdominal aorta and right iliac artery, in serial cross-sections through the LAD and right iliac artery. The surface of the thoracic aorta was mainly covered by non-raised sudanophilic fatty streaking (≈45% surface area), while the abdominal had predominantly raised lesions (≈30% or surface area) (Fig 1C). Lesions in the right iliac artery spanned from fatty streaks to small raised lesions (≈30% in total). Fractional lesion area was not statistically altered by orchiectomy in any of the

vascular beds (Fig 1D–1F). Consistently, average intimal area was not significantly affected by orchiectomy in the LAD and right iliac artery (Fig 1G–1I).

## Castration reduces aggressive behavior in PCSK9$^{D374Y}$ minipigs

Animals where evaluated on aggressive behavior within the last month before euthanization when they were housed in single-pens in stalls with only males. Orchiectomized minipigs exhibited less aggressive behavior than controls as measured by the average score of 10 aggression parameters (Fig 1J). The positive non-aggression parameter (shyness of the observer) was not altered significantly.

## Plaque necrosis in fibroatheromas increased in orchiectomized minipigs

Morphological analysis of lesions in the LAD and right iliac arteries showed mostly early atherosclerotic lesion types consisting of foam cell lesions and pathological intimal thickenings (Fig 2A), and there were no statistically significant differences between groups. To probe for lesion composition at a more advanced stage of atherosclerosis, we did further histological analysis at two sites in the atherosclerosis-susceptible abdominal aorta. A blinded observer harvested the most raised lesion in close proximity to the aortic trifurcation and the most raised lesion in the more proximal part of the abdominal aorta. The majority of lesions were pathological intimal thickenings and fibroatheromas (progressive lesions) and there was no difference in frequency distribution (progressive versus non-progressive nor fibroatheromas versus non-fibroatheromas) between orchiectomized and control animals (Fig 2A). Lesion thickness (a proxy for plaque size) was not affected by castration in the aortic trifurcation nor in the proximal abdominal aortic lesion (Fig 2B). However, analysis of necrotic core formation in the aortic lesions, quantified as the fraction of collagen-free lesion areas in Sirius Red-stained histological sections, revealed a significant higher amount of necrosis in castrated compared with control minipigs (Fig 2C–2E).

## Discussion

In the present study, we tested the effects of perinatal surgical orchiectomy on the development of atherosclerosis in hypercholesterolemic PCSK9$^{D374Y}$ Yucatan minipigs. Orchiectomy led to plasma testosterone at castrate levels defined for human ADT and caused a more atherogenic lipid profile with higher LDL-cholesterol levels similar to what has been reported for hypogonadal men [2]. Atherosclerotic lesion size was not changed in any vascular bed, but at the most atherosclerosis-prone site examined in the abdominal aorta, where fibroatheroma development was frequent at the time point examined, significantly larger necrotic cores were found in castrated minipigs.

The analysis of castration effects fills a knowledge gap in the field of porcine atherosclerosis models. Porcine models have been used for decades for atherosclerosis research, and animals have frequently been castrated for various reasons, but the effects that castration may exert on lesion development has not been addressed. This is not merely a technical point that hinders comparison of different approaches to lesion induction and must be factored into the interpretation of previous pig studies. Effects of castration in experimental models may also point to potential causal mechanisms underlying the association of androgen deprivation therapy, or spontaneous hypogonadism, with coronary events [1,2].

Overall, we found modest effects of castration on plasma lipids and plaque burden in minipigs compared to what has been reported in mice and rabbits [4–8]. As an investigation into the human association of hypogonadism and coronary events, our investigation is unfortunately limited by the scarcity of advanced coronary lesions in the present study. The augmented

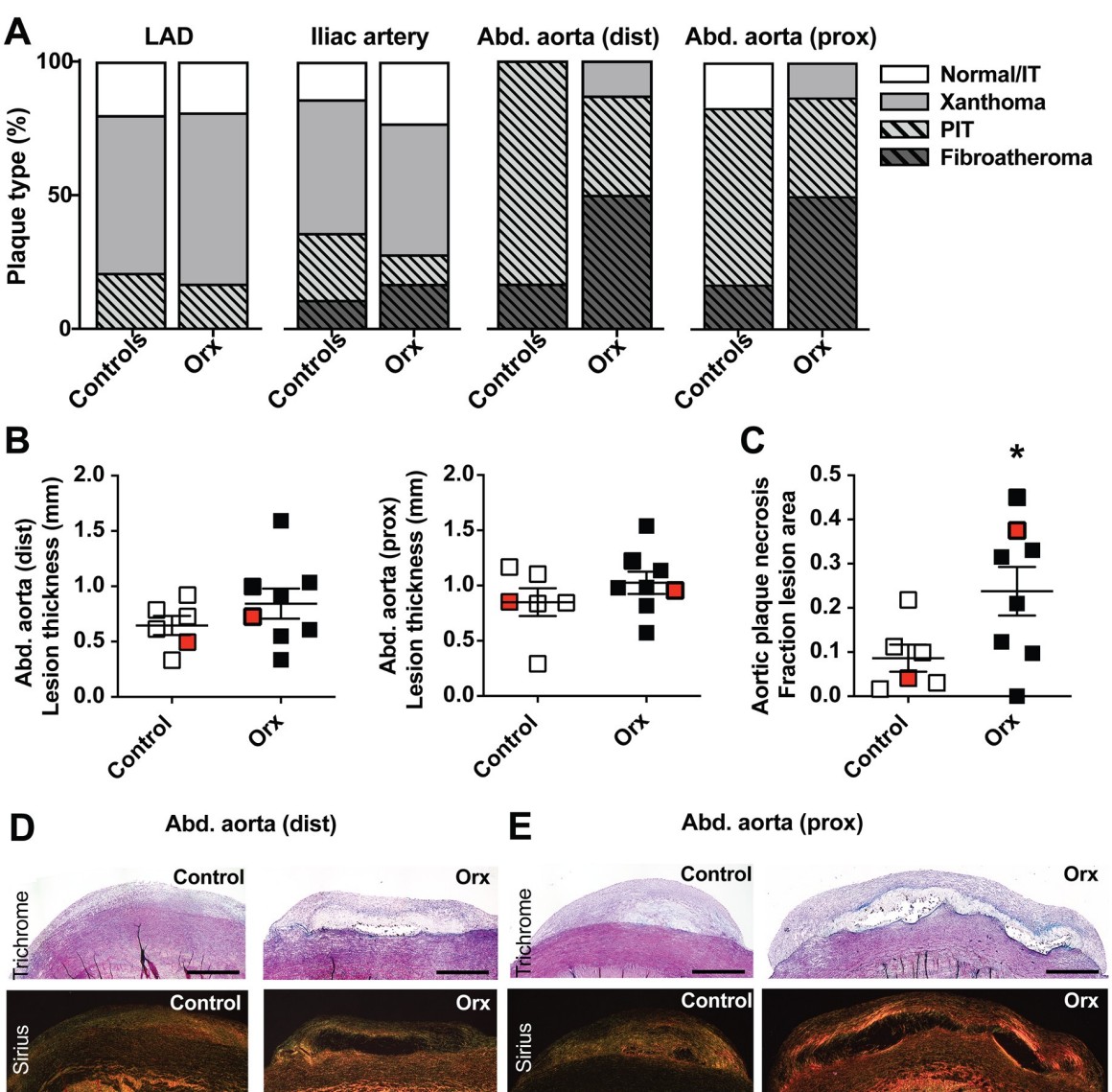

**Fig 2. Plaque typing and core formation. A**. Lesion type (Virmani classification) of sections from the LAD, right iliac and abdominal aorta. For statistical analysis, the most advanced lesion type found in each animal was compared between groups. Differences were not statistically significant (Fisher's exact tests). **B**. Thickness of the most raised lesions in the distal abdominal aorta close to the trifurcation and further proximal were similar between groups. **C**. Plaque necrosis in abdominal aorta lesions (mean of necrosis at the two aortic sites) were significantly increased in castrated animals. *P = 0.036 (t-test with Welch's correction). **D-E**. Representative images (indicated by red dots in B-C) of trichrome- and Sirius Red-stained histological sections of abdominal aorta lesions. Scale bars = 1 mm. Number of data points/animals in each panel are 6 and 8 for the control and orchiectomized groups, respectively. Error bars indicate mean and SEM.

plaque necrosis that we found in the most advanced lesions of the abdominal aorta in castrated minipigs may, however, be interesting in that respect, because it is consistent with observations in advanced murine atherosclerosis. In fibroatheromatous lesions induced by flow-disturbing collars in *Apoe*-knockout mice, Knutson et al. found that castration through GnRH agonism increased necrotic core size without changes in lesion burden [7]. Another study in *Apoe*-knockout mice also demonstrated an androgen receptor-dependent effect of testosterone on necrotic core formation [12]. If castration facilitates necrosis in coronary lesions in humans, this may be a credible mechanism for the relatively rapid onset of the increased risk of heart

attack after ADT [13,14]. It would be interesting in future studies to investigate the short-term effects on plaque necrosis when castration is induced in older PCSK9[D374Y] males with advanced, pre-existing coronary atherosclerosis to test this hypothesis more directly. Such further studies are also warranted since the analysis of abdominal aorta plaque necrosis was an ad hoc finding of the present investigation and not one of the pre-specified analyses.

Although castration in the present study was related to changes in LDL and HDL, the relatively small size of that effect and the absence of changes in plaque burden, indicate that causes of increased plaque necrosis may well be lipid-independent effects. Androgen receptors are present on many cell types in the atherosclerotic lesion, including macrophages, endothelial cells and smooth muscle cells [15]. Previously reported effects of testosterone deficiency, which conceivably could be involved in plaque necrosis, include heightened inflammation by macrophages or T cells [8,16], and reduced reverse cholesterol transport from foam cell macrophages [17]. GnRH receptors are also present on cells partaking in atherosclerotic plaques and may potentially be stimulated by compensatory increases in GnRH levels that result from orchiectomy [7].

The results obtained in the HFHC-fed PCSK9[D374Y] transgenic minipigs may not necessarily extend to all pig models used for atherosclerosis research. Male castration has previously been shown to lead to obesity and decreased insulin sensitivity in adult Göttingen minipigs [18]. It is possible that such metabolic changes contribute to the atherosclerosis phenotype that can be induced in castrated male Göttingen minipigs by high-fat, high-cholesterol diet [19]. The restrictive feeding regimen of the present study may help explain why we observed no changes in body weight and only a moderate increase in backfat thickness in castrated PCSK9[D374Y] transgenic minipigs. Genetic background may also be important, and Yucatan minipigs are known to develop less metabolic changes on high energy diets than Ossabaw minipigs, which are also used for atherosclerosis research [20,21].

Perinatal orchiectomy of boars is standard practice in industrial livestock to prevent a strong undesirable flavor of pork. It also comes with benefits in animal handling and welfare, since entire male pigs exhibit more aggressive behavior causing injuries, especially after sexual maturation. In our work, we confirmed a clear aggression-reducing effect of orchiectomy in PCSK9[D374Y] Yucatan minipigs. Atherosclerosis studies are long term and include extended periods working with sexually mature animals, and animal aggressive behavior is a considerable concern in the work environment for facility staff members. Aggressive behavior is also likely to reduce animal welfare, among other things because it necessitates individual housing of animals. Castration may thus be warranted to improve the working environment as well as animal welfare in preclinical porcine studies of atherosclerosis involving males.

## Study limitations

The relatively small number of animals may have masked our ability to detect some effects of castration on plasma lipids and atherosclerosis traits. Furthermore, as a model to understand effects on atherosclerosis of ADT in humans, pigs like other non-primate models are limited by differences in sex hormone physiology [18]. Another limitation of our study is the lack of advanced coronary atherosclerosis, including thin-cap fibroatheromas that would provide a more direct model of the type of atherosclerosis that may cause excess cardiovascular events in persons with spontaneous hypogonadism or patients undergoing ADT.

## Conclusion

In conclusion, we found orchiectomy to increase plaque necrosis in PCSK9[D374Y] Yucatan minipigs in the absence of changes in atherosclerosis burden and with only a sparse effect on

the atherogenic lipid profile. It also reduced signs of aggression. Together with previous studies in mouse models, our finding points to direct effects of castration on necrotic core formation.

## Supporting information

**S1 Checklist. The ARRIVE guidelines checklist.**
(PDF)

## Acknowledgments

We would like to thank Dorte Qualmann, Lisa Maria Røge, and Carsten Berthelsen for skilled technical assistance.

## Author Contributions

**Conceptualization:** Jeong T. Shim, Charlotte B. Sørensen, Jacob F. Bentzon.

**Data curation:** Jeong T. Shim, Nikolaj Schmidt, Paula Nogales, Torben Larsen, Jacob F. Bentzon.

**Formal analysis:** Jeong T. Shim, Paula Nogales, Torben Larsen, Jacob F. Bentzon.

**Funding acquisition:** Jeong T. Shim, Charlotte B. Sørensen, Jacob F. Bentzon.

**Investigation:** Jeong T. Shim, Nikolaj Schmidt, Paula Nogales, Charlotte B. Sørensen, Jacob F. Bentzon.

**Methodology:** Jeong T. Shim, Charlotte B. Sørensen, Jacob F. Bentzon.

**Project administration:** Jeong T. Shim, Jacob F. Bentzon.

**Resources:** Jacob F. Bentzon.

**Software:** Jeong T. Shim.

**Supervision:** Charlotte B. Sørensen, Jacob F. Bentzon.

**Validation:** Jeong T. Shim, Jacob F. Bentzon.

**Visualization:** Jacob F. Bentzon.

**Writing – original draft:** Jeong T. Shim, Jacob F. Bentzon.

**Writing – review & editing:** Nikolaj Schmidt, Paula Nogales, Torben Larsen, Charlotte B. Sørensen, Jacob F. Bentzon.

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
