## [Decision Letter · Decision Letter 0]

3 Apr 2020

PONE-D-20-06819

Effects of Castration on Atherosclerosis and Aggressive Behavior in Yucatan Minipigs with Genetic Hypercholesterolemia

PLOS ONE

Dear Dr. Bentzon,

Thank you for submitting your manuscript to PLOS ONE. After careful consideration, we feel that it has merit but does not fully meet PLOS ONE’s publication criteria as it currently stands. Therefore, we invite you to submit a revised version of the manuscript that addresses the points raised during the review process.

We would appreciate receiving your revised manuscript by May 18 2020 11:59PM. To enhance the reproducibility of your results, we recommend that if applicable you deposit your laboratory protocols in protocols.io, where a protocol can be assigned its own identifier (DOI) such that it can be cited independently in the future. For instructions see: http://journals.plos.org/plosone/s/submission-guidelines#loc-laboratory-protocols

We look forward to receiving your revised manuscript.

Kind regards,

Michael Bader

Academic Editor

PLOS ONE

Journal Requirements:

1. As part of your revision, please complete and submit a copy of the ARRIVE Guidelines checklist, a document that aims to improve experimental reporting and reproducibility of animal studies for purposes of post-publication data analysis and reproducibility: https://www.nc3rs.org.uk/arrive-guidelines. Please include your completed checklist as a Supporting Information file. Note that if your paper is accepted for publication, this checklist will be published as part of your article.

2. We note that you have a patent relating to material pertinent to this article. Please provide an amended statement of Competing Interests to declare this patent (with details including name and number), along with any other relevant declarations relating to employment, consultancy, patents, products in development or modified products etc. Please confirm that this does not alter your adherence to all PLOS ONE policies on sharing data and materials, as detailed online in our guide for authors http://journals.plos.org/plosone/s/competing-interests by including the following statement: "This does not alter our adherence to  PLOS ONE policies on sharing data and materials.” If there are restrictions on sharing of data and/or materials, please state these. Please note that we cannot proceed with consideration of your article until this information has been declared.

Reviewers' comments:

Reviewer's Responses to Questions

**Comments to the Author**

1. Is the manuscript technically sound, and do the data support the conclusions?

Reviewer #1: Yes

Reviewer #2: Partly

2. Has the statistical analysis been performed appropriately and rigorously? 

Reviewer #1: Yes

Reviewer #2: I Don't Know

3. Have the authors made all data underlying the findings in their manuscript fully available?

Reviewer #1: Yes

Reviewer #2: Yes

4. Is the manuscript presented in an intelligible fashion and written in standard English?

Reviewer #1: Yes

Reviewer #2: Yes

5. Review Comments to the Author

Reviewer #1: Shim et al. have studied the effects of neonatal castration on atherosclerosis and aggressive behavior in yucatan minipigs with genetic hypercholesterolemia. The study is well performed and adequately discussed.

I have only a few minor points.

1. Background, page 4, line 10: Remove the words ”through changes in T cell maturation”.

2. Were any samples below LOD of the testosterone assay?

3. Insert a reference or a more detailed description of Virmani classification of plaque morphology.

4. Replace the rather old reference 3 with a recent review (Arterioscler Thromb Vasc Biol. 2020 Mar;40(3):e55-e64. doi: 10.1161/ATVBAHA.119.313046. Epub 2020 Jan 23. Cardiovascular Effects of Androgen Deprivation Therapy in Prostate Cancer: Contemporary Meta-Analyses.)

5. Discussion: Not only GnRH agonist but also direct testosterone/AR-mediated modulation of necrotic core has been shown in mouse models (Bourghardt et al. Endocrinology. 2010 Nov;151(11):5428-37).

6. The authors mention species differences in sex hormone physiology. Which are they? Clarify or at least insert a reference.

Reviewer #2: Letter of comments for authors

The authors present a paper, evaluating the effect of castration on atherogenesis and behavior in Yucatan minipigs. The content of the paper is overall interesting and of relevance from a model development point of view; in particular considering atherosclerosis studies in pigs being costly and time consuming where standard approaches done for e.g. practical reasons (such as castration), may have an impact on the disease progression. The data presented include lipid levels, histology incl. morphology and morphometric measures in both aorta and LAD. The data is presented well and accurately in the figures and pictures. Additionally, the animals have been observed with regards to behavior before and after castration.

The conclusion of the study is that modest effects of castration is observed on plasma lipids and plaque burden in minipigs.

The text presents with some deficiencies in terms of method description and aspects discussed. Additionally, the text should be revised with regards to wordings, grammatic and in general be more concise and accurate. Also check that referred methods software and systems are described according to guidelines of the journal.

Recommendation: For publication the paper should go through a major review in order to focus the aim, discussion and perspective of the study.

Specific inquires:

Title: Suggestion that the title should cover the primary focus of the study – is this to evaluate effects on behavior or atherogenesis? Proposal to narrow this down to the latter and include the behavioral aspect to be as part of perspective of the model. Also, the conclusion only touches on the atherogenesis part, supporting to narrow the main scope of the study.

Abstract: Please include up-front the number of animals evaluated as well as age and duration of study, for the reader to quickly understand what the results show.

Introduction: Although it may be correct that effect of neutralization has not been systematically evaluated in pig models, it should be taken into consideration and referred that effects on metabolic syndrome and obesity has been evaluated systematically in Göttingen Minipigs (e.g. Christoffersen BO Steroids. 2010 Oct;75(10):676-84. doi:10.1016/j.steroids. 2010.04.004. – other more novel papers may refer to this).

Methods: Please provide more details on orchiectomy: Was the anesthesia per animal or per weight? Local anesthesia? How were they monitored? Any posttreatment analgesia? Can be in supplementary section.

Please provide details as to feeding and dietary composition as pathology is driven by diet.

Please provide reference as to assessment scheme for behavioral observations. Is this a standard assessment in the species?

Please provide details to observer: is this a person the animals are familiar with or not?

Please state how “shyness” is scored. This is a subjective term

Lipoprotein fraction data is presented but method of assessment not specified. Please include this methodology.

It is apparent from reading the result section that evaluation of aortic lesions was done to ensure advanced lesions – however it is a little confusing from the method section that this is what has been done. The authors should clarify the following section in methods to avoid confusion.

“Two plaques in the abdominal aortas were also obtained: One distal, transversal segment containing the most raised lesion (by macroscopic inspection) in the region close to the aortic trifurcation, and one proximal segment containing the most raised lesion further proximal in the abdominal aorta.”

Provide reference to “Lesions were categorized according to the Virmani classification as normal, xanthoma, pathological intimal thickening or fibroatheroma.”

For statistical analyses: Has data been tested for normal distribution? Consider executing a repeated measure multivariate analyses instead of t-test of AUC and include LDL-C as explanatory variable to address whether observed differences in necrotic core could be explained by LDL cholesterol differences, although only subtle differences are observed.

Results:

Figures: Well-illustrated figures and figures and in particular very useful to show all data points (in the unpaired data, where it makes sense) including from what individual histology is depicted.

Figure 1 legend: Please include what histological staining that is used H-I.

Text: If direct measures of body composition specifically fat% is available from these animals, this would be highly relevant.

For the headline please rephrase the word “facilitate” (“Castration facilitates plaque necrosis in fibroatheromas”).

Discussion: Again, as stated for the introduction, previous studies in minipigs should be taken into consideration. Additionally, beside the studies exploring effects of castration, discussion of breed susceptibility could also be taken in, with reference to the e.g. Neeb et al suggesting the Yucatan to be less prone to metabolic syndrome compared to Ossabaw pigs (Neeb Z et al Comp Med. 2010 Aug; 60(4): 300–315. PMID: 20819380). The finding of no difference in bodyweight between the two groups is somewhat surprising.

Rephrase “…castration facilitates atherosclerosis..” – the data presented suggests an association between castrated animals and increased necrotic core, but actual mechanistic studies to cover the underlying biology has not been executed.

6. PLOS authors have the option to publish the peer review history of their article (what does this mean?). If published, this will include your full peer review and any attached files.

Reviewer #1: No

Reviewer #2: No

---

## [Author Response · Author response to Decision Letter 0]

6 May 2020

Editor comments

1. As part of your revision, please complete and submit a copy of the ARRIVE Guidelines checklist, a document that aims to improve experimental reporting and reproducibility of animal studies for purposes of post-publication data analysis and reproducibility: https://www.nc3rs.org.uk/arrive-guidelines. Please include your completed checklist as a Supporting Information file. Note that if your paper is accepted for publication, this checklist will be published as part of your article.

Response: We have filled the ARRIVE Guidelines checklist and expanded on our description of animals and data analysis throughout the methods section. 

2. We note that you have a patent relating to material pertinent to this article. Please provide an amended statement of Competing Interests to declare this patent (with details including name and number), along with any other relevant declarations relating to employment, consultancy, patents, products in development or modified products etc. Please confirm that this does not alter your adherence to all PLOS ONE policies on sharing data and materials, as detailed online in our guide for authorshttp://journals.plos.org/plosone/s/competing-interests by including the following statement: "This does not alter our adherence to PLOS ONE policies on sharing data and materials.” If there are restrictions on sharing of data and/or materials, please state these. Please note that we cannot proceed with consideration of your article until this information has been declared.

Response: We have the patent on the D374Y-PCSK9 minipigs. The existence of this patent does not alter our adherence to PLOS ONE policies on sharing data and materials, which is now also stated in text. 

Reviewer #1 comments

Shim et al. have studied the effects of neonatal castration on atherosclerosis and aggressive behavior in yucatan minipigs with genetic hypercholesterolemia. The study is well performed and adequately discussed.

Response: We thank the reviewer for the time and effort invested in reading and suggesting improvements and clarifications for our manuscript.

I have only a few minor points.

1. Background, page 4, line 10: Remove the words ”through changes in T cell maturation”.

Response: We have now removed this part of the sentence and we thank the reviewer for helping us making the point in this sentence clear. 

2. Were any samples below LOD of the testosterone assay?

Response: We detected levels of testosterone in all samples; the lowest value we measured was 68.3 pg/ml (LOD was lower or equal to 40 pg/ml).

3. Insert a reference or a more detailed description of Virmani classification of plaque morphology.

Response: In the final sentence in the Methods section, we have now added the reference to the original paper by Virmani et al (reference 11) which contains a detailed description of the Virmani classification.

4. Replace the rather old reference 3 with a recent review (Arterioscler Thromb Vasc Biol. 2020 Mar;40(3):e55-e64. doi: 10.1161/ATVBAHA.119.313046. Epub 2020 Jan 23. Cardiovascular Effects of Androgen Deprivation Therapy in Prostate Cancer: Contemporary Meta-Analyses.)

Response: We thank the reviewer for pointing out this new paper as an up-to-date reference. We have replaced this reference.

5. Discussion: Not only GnRH agonist but also direct testosterone/AR-mediated modulation of necrotic core has been shown in mouse models (Bourghardt et al. Endocrinology. 2010 Nov;151(11):5428-37).

Response: Thank you for correcting our omission of this important observation. We have referred to this work in the discussion of the revised manuscript (lines 247-248 and ref 12). 

6. The authors mention species differences in sex hormone physiology. Which are they? Clarify or at least insert a reference.

Response: We have chosen to omit the part of the sentence in the introduction stating this. For the limitation sections we have added reference to Christoffersen B et al (ref 18) in which differences in estradiol/testosterone ratios between pigs and human are described. 

 

Reviewer #2 comments

The authors present a paper, evaluating the effect of castration on atherogenesis and behavior in Yucatan minipigs. The content of the paper is overall interesting and of relevance from a model development point of view; in particular considering atherosclerosis studies in pigs being costly and time consuming where standard approaches done for e.g. practical reasons (such as castration), may have an impact on the disease progression. The data presented include lipid levels, histology incl. morphology and morphometric measures in both aorta and LAD. The data is presented well and accurately in the figures and pictures. Additionally, the animals have been observed with regards to behavior before and after castration.

The conclusion of the study is that modest effects of castration is observed on plasma lipids and plaque burden in minipigs.

The text presents with some deficiencies in terms of method description and aspects discussed. Additionally, the text should be revised with regards to wordings, grammatic and in general be more concise and accurate. Also check that referred methods software and systems are described according to guidelines of the journal.

Recommendation: For publication the paper should go through a major review in order to focus the aim, discussion and perspective of the study.

Response: We thank the reviewer for the time and effort invested in reading and suggesting improvements and clarifications for our manuscript.

Specific inquires:

Title: Suggestion that the title should cover the primary focus of the study – is this to evaluate effects on behavior or atherogenesis? Proposal to narrow this down to the latter and include the behavioral aspect to be as part of perspective of the model. Also, the conclusion only touches on the atherogenesis part, supporting to narrow the main scope of the study.

Response: Thank you for this suggestion. We agree that this could be a good idea and have changed the title to: “Effects of Castration on Atherosclerosis in Yucatan Minipigs with Genetic Hypercholesterolemia“.

Abstract: Please include up-front the number of animals evaluated as well as age and duration of study, for the reader to quickly understand what the results show.

Introduction: Although it may be correct that effect of neutralization has not been systematically evaluated in pig models, it should be taken into consideration and referred that effects on metabolic syndrome and obesity has been evaluated systematically in Göttingen Minipigs (e.g. Christoffersen BO Steroids. 2010 Oct;75(10):676-84. doi:10.1016/j.steroids. 2010.04.004. – other more novel papers may refer to this).

Response: Thank you for suggesting these improvements. We have inserted the number of animals as well as more information on experimental design. We have also rewritten the results section for clarity. The mention of other relevant endpoints affected by castration, including metabolic syndrome, has been incorporated in the discussion (lines 263-271)

Methods: Please provide more details on orchiectomy: Was the anesthesia per animal or per

weight? Local anesthesia? How were they monitored? Any posttreatment analgesia? Can be

in supplementary section.

Response: Orchiectomy was performed by a standard agricultural procedure with anesthesia in the facility that breeds our transgenic minipigs using a long-acting NSAID (flunixine meglumine). Anesthesia was per animal and the piglets were monitored by the stable staff afterwards. No posttreatment analgesia was given. Some animals outside those described in the paper were excluded from the study and euthanized because of the development of scrotal hernia (n=3). These hernias were occurring generally among the male pigs in the breeding colony at the time. After selecting other breeders from our colony for subsequent studies, we are seeing this rarely, including in castrated pigs. We therefore believe that it is genetic and not related to the procedure. In the present study it was seen in both castrated (n=2) and non-castrated pigs (n=1). We have edited the text to describe this. It can be found on lines 100-102 and lines 110-115). 

Please provide details as to feeding and dietary composition as pathology is driven by diet.

Response: We have inserted more details about the diet and feeding regimen on lines 105-109. Pigs were fed ad libitum until they reached 20-25 kg of body weight and was thereafter fed 700 g of feed divided in two daily portions. 

Please provide reference as to assessment scheme for behavioral observations. Is this a standard assessment in the species?

Response: This was an assessment scheme we developed in collaboration with the animal caretakers. To our knowledge there is no standard assessment scheme for pig aggressive behavior. We have inserted more detail about the assessment on lines 118-130. 

Please provide details to observer: is this a person the animals are familiar with or not?

Response: The persons scoring the pigs were part of the daily animal staff and familiar to the pigs. We have added that information to lines 125-126. 

Please state how “shyness” is scored. This is a subjective term

Response: “Shyness” was used to describe the behavior of pigs attempting to keep a distance to the observer at all times. This information has been added to lines 124-125.

Lipoprotein fraction data is presented but method of assessment not specified. Please include this methodology.

Response: LDL and HDL cholesterol were measured by the direct method using the standard assays for the Siemens ADVIA1800 autoanalyzer. This has now been added to the manuscript on lines 133-136. 

It is apparent from reading the result section that evaluation of aortic lesions was done to ensure advanced lesions – however it is a little confusing from the method section that this is what has been done. The authors should clarify the following section in methods to avoid confusion.

“Two plaques in the abdominal aortas were also obtained: One distal, transversal segment containing the most raised lesion (by macroscopic inspection) in the region close to the aortic trifurcation, and one proximal segment containing the most raised lesion further proximal in the abdominal aorta.”

Response: As the reviewer notes, our goal was to look at the most advanced lesions in the abdominal aorta, which are also the most advanced lesions overall. We have rephrased the sentence so that it now reads: “The most advanced lesion (by macroscopic inspection) in two regions of the abdominal aorta were also obtained. Lesions develop reproducibly in the distal aorta near the aortic trifurcation and a cross-sectional slice containing the most raised lesion in this region was selected. Furthermore, a cross-sectional slice containing the most raised lesion furthermore proximal in the abdominal aorta (typically near the renal arteries) were obtained”. The revised text can be found on lines 157-165. 

Provide reference to “Lesions were categorized according to the Virmani classification as normal, xanthoma, pathological intimal thickening or fibroatheroma.”

Response: In the final sentence in the Methods section (lines 168-169), we have now added the reference to the original paper by Virmani et al (reference 11) which contains a detailed description of the Virmani classification.

For statistical analyses: Has data been tested for normal distribution? Consider executing a repeated measure multivariate analyses instead of t-test of AUC and include LDL-C as explanatory variable to address whether observed differences in necrotic core could be explained by LDL cholesterol differences, although only subtle differences are observed.

Response: Yes, data was tested for normal distribution with the Shapiro-Wilk normality test and this information is now included in the statistics section on lines 172-180. We appreciate the suggestion to do repeated measures ANOVA on the plasma lipid data. The reason that we prefer the AUC approach is that this type of statistics fits well with the biological understanding of the connection between plasma lipids and atherosclerosis, i.e. that it is the cumulative exposure that is important for the amount of atherosclerosis. We did not find any significant correlation between LDL cholesterol (measured as area-under-the-curve over the course of study) and plaque necrosis: 

Results:

Figures: Well-illustrated figures and figures and in particular very useful to show all data

points (in the unpaired data, where it makes sense) including from what individual histology is depicted.

Figure 1 legend: Please include what histological staining that is used H-I.

Response: We have added that the images in panel G are stained with hematoxylin-eosin, and that H-I is data measured on hematoxylin-eosin-stained sections. 

Text: If direct measures of body composition specifically fat% is available from these animals, this would be highly relevant.

Response: We appreciate this suggestion, but unfortunately, we do not have that information. 

For the headline please rephrase the word “facilitate” (“Castration facilitates plaque necrosis in fibroatheromas”).

Response: We appreciate this advice and have rephrased to “Plaque necrosis increased in fibroatheromas of orchiectomized pigs”

Discussion: Again, as stated for the introduction, previous studies in minipigs should be taken into consideration. Additionally, beside the studies exploring effects of castration, discussion of breed susceptibility could also be taken in, with reference to the e.g. Neeb et al suggesting the Yucatan to be less prone to metabolic syndrome compared to Ossabaw pigs (Neeb Z et al Comp Med. 2010 Aug; 60(4): 300–315. PMID: 20819380). The finding of no difference in bodyweight between the two groups is somewhat surprising.

Response: We have added a section discussing the effect of castration on metabolic parameters in pigs and the effect of breed on lines 263-271. In that context we have also discussed the importance of ad libitum versus restricted feeding. 

Rephrase “…castration facilitates atherosclerosis..” – the data presented suggests an association between castrated animals and increased necrotic core, but actual mechanistic studies to cover the underlying biology has not been executed.

Response: We agree with the reviewer and have changed the headline in the results section. I do not think that we make that statement about our data in any other parts of the manuscript.

---

## [Decision Letter · Decision Letter 1]

20 May 2020

Effects of Castration on Atherosclerosis in Yucatan Minipigs with Genetic Hypercholesterolemia

PONE-D-20-06819R1

Dear Dr. Bentzon,

We are pleased to inform you that your manuscript has been judged scientifically suitable for publication and will be formally accepted for publication once it complies with all outstanding technical requirements.

With kind regards,

Michael Bader

Academic Editor

PLOS ONE

Additional Editor Comments (optional):

Reviewers' comments:

Reviewer's Responses to Questions

**Comments to the Author**

1. If the authors have adequately addressed your comments raised in a previous round of review and you feel that this manuscript is now acceptable for publication, you may indicate that here to bypass the “Comments to the Author” section, enter your conflict of interest statement in the “Confidential to Editor” section, and submit your "Accept" recommendation.

Reviewer #2: All comments have been addressed

2. Is the manuscript technically sound, and do the data support the conclusions?

Reviewer #2: Yes

3. Has the statistical analysis been performed appropriately and rigorously? 

Reviewer #2: Yes

4. Have the authors made all data underlying the findings in their manuscript fully available?

Reviewer #2: Yes

5. Is the manuscript presented in an intelligible fashion and written in standard English?

Reviewer #2: Yes

6. Review Comments to the Author

Reviewer #2: (No Response)

7. PLOS authors have the option to publish the peer review history of their article (what does this mean?). If published, this will include your full peer review and any attached files.

Reviewer #2: No

---

## [Editor Report · Acceptance letter]

27 May 2020

PONE-D-20-06819R1 

Effects of Castration on Atherosclerosis in Yucatan Minipigs with Genetic Hypercholesterolemia 

Dear Dr. Bentzon:

I am pleased to inform you that your manuscript has been deemed suitable for publication in PLOS ONE. Congratulations! Your manuscript is now with our production department. 

With kind regards,

on behalf of

Prof. Michael Bader 

Academic Editor

PLOS ONE